# Qualitative and Quantitative Ethnobotanical Survey in Al Baha Province, Southwestern Saudi Arabia

Sami Asir Al-Robai [1,*], Aimun A. E. Ahmed [2,3], Haidar Abdalgadir Mohamed [1,4], Abdelazim Ali Ahmed [1,5], Sami A. Zabin [6] and Abdullah A. A. Alghamdi [1]

1  Department of Biology, Faculty of Science, Albaha University, Al Baha 65431, Saudi Arabia
2  Pharmacology Department, Faculty of Medicine, Albaha University, Al Baha 65431, Saudi Arabia
3  Pharmacology Department, Faculty of Pharmacy, Omdurman Islamic University, Khartoum 11111, Sudan
4  Medicinal and Aromatic Plants Research Institute, National Center for Research, Khartoum 11111, Sudan
5  Department of Botany, Faculty of Science, University of Khartoum, Khartoum 11115, Sudan
6  Department of Chemistry, Faculty of Science, Albaha University, Al Baha 65431, Saudi Arabia
*  Correspondence: dr.alrobai@gmail.com; Tel.: +966556616866

**Abstract:** The documentation of ethnobotanical knowledge is useful for biocultural conservation, preserving the diversity of plants, and drug development. The present study was carried out to compile and document the knowledge and uses of plants in Al Baha Province, Kingdom of Saudi Arabia (KSA). A total of 81 knowledgeable informants of different sexes, ages, and status levels were randomly selected and interviewed. The majority of the informants (63%) were >60 years old. The collected data were qualitatively and quantitatively described using different quantitative indices [family importance value (FIV), use value (UV), and informant consensus factor (ICF)]. The fidelity level (FL), rank order priority (ROP), and relative popularity level (RPL) were determined for the plants (42 species) mentioned by at least three informants. A total of 97 plants belonging to 91 genera and 44 families were reported. The most commonly used parts were fruits (30.7%) and leaves (25.4%), whereas the most frequently used modes of preparation were maceration (24.7%) and direct application (23.3%). Most of the cited plants (49.5%) were traditionally used for gastrointestinal tract (GIT) disorders, whereas a few plants (6.2%) were used for the treatment of reproductive disorders. The most ethnobotanically important families were Lamiaceae (FIV = 0.35) and Apiaceae (FIV = 0.33). The highest UV was represented by *Zingiber officinale* (0.086) followed by *Commiphora myrrha* and *Trigonella foenum-graecum* (0.074). The level of agreement among the interviewees was remarkably high (ICF = 0.65–0.93) for plants that had the ability to cure infectious diseases. A low level of agreement (ICF = 0.33–0.48) was observed among the informants towards plants that were used to treat gastrointestinal tract, reproductive, hematological, and central nervous system disorders. There was a total and absolute disagreement (ICF = 0) among the informants regarding the plants that were used to treat renal, endocrine system, oncological/immunological, rheumatic, orthopedic, ear, nose, and throat (ENT), and inflammatory disorders. Six of the plants which were cited by three informants or more had a high healing efficacy (FL = 100) and forty species attained ROP values of 50 or above. Out of the 42 plants, 20 species were grouped as popular (RPL = 1), and the remaining plants (22 species) were unpopular (RPL < 1). *Curcuma longa*, which showed the highest ROP value (100), was used to enhance immunity. In conclusion, various plant species in Al Baha province were used by the local communities for the treatment of different health problems. The documentation of these plants could serve as a basis for further scientific research and conservation studies.

**Keywords:** Al Baha province; ethnobotany; traditional knowledge; quantitative indices; medicinal plants



## 1. Introduction

Since the beginning of life, plants have been used as food, forage, and considered a good source of medications, dyes, cosmetics, fibers, etc. [1]. Ethnobotany plays a crucial role

in the exploration, documentation, and preservation of traditional indigenous knowledge within the community. It plays an important role in documenting and correlating social and indigenous knowledge with the healing potential of various ailments, thus constituting a platform for plant-derived drug discovery [2–5]. In addition, documentation of the traditional indigenous knowledge about the medicinal values of plant species resulted in the development of a number of vital modern drugs [6].

Before the occurrence of modern drugs and their application in the health care system, people utterly depended on traditional medicine [7]. According to the WHO, 80% of the world's population relies on traditional medicinal practitioners for their healthcare needs, which reflects the importance of traditional medicine, particularly in developing countries [8]. Plants constitute a complex traditional system of medicine and are considered a cultural heritage [9]. Serendipitous exploration of the medicinal potentials of certain plants can be conducted through animal utilization and consumption of such plants for the healing purposes of various ailments. This biorational could possibly lead to the discovery of novel secondary phytoconstituents [9].

Recently, great attention has been paid to natural resources such as microbes, animals, plants, and marine organisms due to their potential role as novel sources for promising drug discovery or therapeutic agents [10–13]. They provided different bioactive constituents contributing to various biological activities [14,15]. Numerous medicinal or nutritious plant constituents are not only used as curative agents, but also as preventive remedies for several harmful chronic diseases [16]. Moreover, natural products may be more reliable, safe, and affordable than synthetic drugs, which may have adverse effects [17]. The safety of herbal remedies concerns both national health authorities and the general public. The traditional utilization of medicinal plants refers to its long historical uses. Their use is well-recognized and widely acknowledged as safe, effective, and frequently accepted by national authorities [18]. By incorporating quantitative research methods in data collection, processing, and interpretation of ethnobotanical results, a growing interest in improving traditional ethnobotanical studies may be achieved [19].

The Kingdom of Saudi Arabia (KSA) is a vast arid land in the Arabian Peninsula that covers approximately 2,250,000 km$^2$. It is roughly located between the latitudes of $15°45'$ N and $34°35'$ N and the longitudes of $34°40'$ E and $55°45'$ E [20,21]. Variable environmental factors such as topography, geomorphology, climate, and soil reveal distinctive ecological habitats, vegetation zones, and, thus, rich flora. It is endowed with a wide range of ecosystems and biodiversity, particularly in the southwestern region [22]. The diversity of the flora of Saudi Arabia provides a remarkably rich source of agricultural and medicinal plants [23]. Saudi Arabian flora is similar to the plants found in East Africa, North Africa, the Mediterranean, and the Irano-Turanian countries in the Saharo-Sindian or Saharo-Arabian region (Holarctic origin) [24,25].

As a result of the inheritance of ethnobotanical knowledge from one generation to another, the Arab regions have a rich inventory of natural medicinal herbs [26]. From ancient times, Arabs had great perceptions and skills in diagnosing and treating various ailments [27]. Traditional medicine is a significant aspect of Saudi Arabia's heritage and was broadly used before the existence of biomedicine [28,29]. Accordingly, more than 1200 plants from Saudi Arabia's flora are of medicinal value [30,31]. Saudis use medicinal plants as traditional remedies to heal various human and livestock diseases [32]. Medicinal plants used in the Kingdom of Saudi Arabia (KSA) have been documented in two volumes, the "Medicinal Plants of Saudi Arabia", published in 1987 and 2000 [33,34]. Several ethnobotanical and ethnomedicinal studies in Saudi Arabia have been conducted [23,32,35–38]. Al-Said [39] described twenty plant species traditionally used in Saudi Arabia with their main chemical constituents. Moreover, an ethnopharmacological survey conducted by Ali et al. [40] showed that 39 plant taxa belonging to 28 plant families are used for treating more than 20 types of ailments.

Al Baha province has a unique location among the Al-Sarwaat mountains with different climates that vary from coastal plain regions up to high altitude mountains that have

a high floristic diversity. However, such floristic diversity and the traditional uses of the local plants are not yet well documented. The present study was carried out to document the ethnobotanical knowledge of Al Baha local communities using both qualitative and quantitative methods.

## 2. Materials and Methods

### 2.1. Study Area

Al Baha province is located between latitudes 20°10′ and 20°15′ N, longitudes 41°15′ and 41°20′ E, and elevations ranging from 260 to 2450 m.a.s.l. It is a part of the Al-Sarawat mountain chain, which is characterized by coarse pink granite, mixed with grey diorite and granodiorite [41]. This area has many steep rocky mountains, hills, steppes, coastal land, and several valleys that contain water during rainy seasons. Roughly, the study area can be classified as dry highland, wet highland, and hot coastal land. Its flora is a mixture of the tropical African and Sudanian plant geographical regions (Paleotropical origin) with very few of the Saharo-Sindian or Saharo-Arabian regions (Holarctic origin) and Mediterranean regions [25]. The study covered ten districts (Al-Qara, Al-Agig, Al-Mandag, Al-Hajrah, Bani Hassan, Al-Baha, Qilwah, Biljurashi, Al-Mukhwah, and Gamid Alzenad) of the province (Figure 1). The residents of the study area are distinguished by homogeneity and social conformity. The population's lives are based on Islam and their habits and customs are considered as mandatory law. Islamic values control the relations within the society [42].

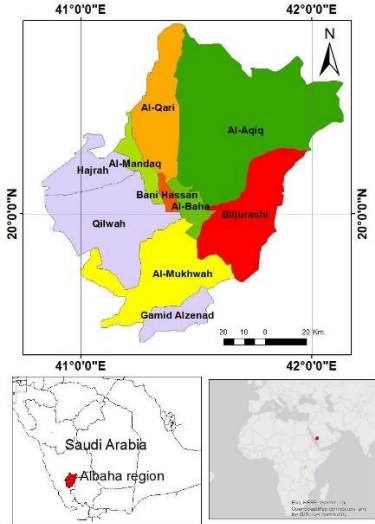

**Figure 1.** Al Baha province map shows the location of the studied districts.

### 2.2. Plant Information, Data Collection and Documentation

The study was conducted during the period from November 2020 to January 2022. The data collection was conducted using pre-prepared semi-structured interviews, group discussion, and informal discussion between the informants and the interviewers. A total of 81 knowledgeable informants of both genders (55 males and 26 females), including expert herbalists, were interviewed. The interviews focused on the local names of the plants used, their traditional uses, parts of the plants used, preparation methods, and administration routes for treating specific ailments. Most of the informants were male (67.9%), non-governmental (84%), and aged above 60 years old (63%) (Table 1). The dominance of male informants in this survey could be attributed to the fact that women in traditional and rural communities do not speak to male foreigners. The informant's demographic data, plant vernacular names, plant parts used, preparation modes, and local traditional uses of the plants were recorded on the spot.

**Table 1.** Demography of interviewees, N = 81.

| Category | Subcategory | Frequency | Percentage % |
| --- | --- | --- | --- |
| **Gender** | Male | 55 | 67.9 |
| | Female | 26 | 32.1 |
| **Age** | <20 | 2 | 2.5 |
| | 21–40 | 9 | 11 |
| | 41–60 | 19 | 23.5 |
| | >60 | 51 | 63 |
| **Status** | Governmental | 13 | 16 |
| | Non-governmental | 68 | 84 |

*2.3. Botanical Identification*

The cited plant species were identified and taxonomically classified by an expert taxonomist (Dr. Haidar A. Mohamed, Department of Biology, Faculty of Science, Al Baha University) and checked with herbarium materials and various volumes of the flora of Saudi Arabia [43–49]. Current databases [50,51] were consulted for name verifications. Sample specimens were deposited at Department of Biology, Faculty of Science, Al Baha University.

*2.4. Data Analysis and Quantitative Indices*

The recorded ethnobotanical knowledge about the reported plants in the study area was analyzed and summarized using descriptive methods. Plant Latin names, family, vernacular names, plant parts used, mode of preparation, ethnobotanical uses, and use value (UV) of each plant were recorded. The preparation methods for the different parts used were precisely noted. Traditional ethnobotanical uses were grouped and categorized as follows: (A) systemic disorders include: gastrointestinal tract (GIT) disorders [diarrhea–constipation–antispasmodic–carminative–abdominal cramps–nausea and vomiting (NV)–anorexia–ulcerative colitis and irritable bowel syndrome (IBS)–hepatoprotective–gall bladder–dyspepsia], cardiovascular disorders [hypertension (HTN)–coronary artery disease (CAD)–myocardial infarction (MI)–edema–cardioprotective], central nervous system/mental and behavioral disorders [mental–headache and migraine–anorexia–cognitive enhancer–vertigo–Alzheimer's–bipolar], hematological disorders [lowering cholesterol levels–anti-coagulant–blood purification–hemoglobin (Hb) and anemia–coagulant–circulatory stimulator], renal disorders [renal function improvement–diuretic–renal stone–urinary tract infection (UTI)–urinary retention], respiratory disorders [ventilation enhancer–cough and expectorant–common cold and flu–asthma], endocrine disorders [diabetes mellitus (DM)–thyroid–acne] and reproductive disorders [prostate tumor–male fertility–prostate–dysmenorrhea–lactogogue]. (B) Non-systemic disorders include: skin disorders [cosmetic applications and various skin diseases], oncological/immunological, rheumatic, orthopedic, inflammatory, ear, nose, and throat (ENT) and ophthalmic disorders. (C) Infectious diseases include: bacterial, viral, fungal, and parasitic diseases. (D) Miscellaneous applications include: dental applications, toxicological effects, insecticidal/repellents, wound healing, nutrition/supplements and weight reduction.

2.4.1. Family Importance Value (FIV)

This index was calculated using the reported formula shown below:

$$FIV = \frac{FC}{N} \times 100$$

where FC is the number of individuals citing the specific family and N stands for the total number of informants. High FIV values indicate great awareness and low values indicate weak awareness of particular plant families [52].

### 2.4.2. Use Value (UV)

Use value (UV) is used to estimate all the possible uses of a plant taxon. The value was calculated using the formula below [53,54]:

$$UV = \frac{\sum Ui}{N}$$

where (Ui) is the number of informants citing various uses of a certain plant taxon, and (N) is the total number of informants (N = 81). A high use value indicates the potential significance of the mentioned plant species.

### 2.4.3. Informant Consensus Factor (ICF)

An informant consensus factor (ICF) was calculated to find out the agreement level among informants for plant species used to treat a particular element category [55]. ICF is calculated using the following formula:

$$ICF = \frac{Nur - Nt}{Nur - 1}$$

where "Nur" is the number of use reports for a particular use category and "Nt" is number of taxa used to treat that particular use category by informants. A high ICF value indicates the high agreement level among informants on the specific plant used for treating a specific use category, whereas a low ICF value indicates a low level of agreement among the informants.

### 2.4.4. Fidelity Level (FL%)

Fidelity level (FL%) is used to assess the importance of individual species in each group to prefer one plant over another [56]. The fidelity level (FL%) was calculated using the following formula:

$$FL = \frac{Ip}{Iu} \times 100$$

where "Ip" is the number of informants who independently suggested the use of a species for the same major purposes, while "Iu" is the total number of informants who mentioned the plant for any use. A high FL value indicates high use of the plant for treating a particular ailment.

### 2.4.5. Relative Popularity Level (RPL)

Relative popularity level (RPL) is a ratio between the number of the major use reports mentioned for a specific taxon and the number of interviewees who cited that taxon for any use reports. RPL value ranges from 0–1; one indicates total popularity and zero indicates unpopularity. All the encountered plant species were divided into popular (RPL = 1) and unpopular (RPL < 1) groups [57].

### 2.4.6. Rank Order Priority (ROP)

The rank order priority or correct value of FL was calculated by multiplying RPL value by FL value (ROP = RPL × FL) [56,58]. A high ROP value indicates the high potential of the plant. It is useful for exploring the most popular species that are used to treat a particular disease. It could be useful for screening plants that have bioactive agents.

## 3. Results and Discussion

The present study indicated that the inhabitants of Al Baha province have their own indigenous ethnobotanical knowledge with a diverse use of plants. It is clear that most of the local people in this province rely mainly on wild, cultivated, and exotic plants for their daily sustenance and the treatment of a variety of ailments. The recorded information confirmed that local traditional knowledge about plant use continues to exist in

various communities in Saudi Arabia. Most of the uses and ethnobotanical knowledge reported were mentioned by elderly people, and men had more traditional ethnobotanical knowledge than women. Indeed, elderly people spend a lot of time with their ancestors, who have a great knowledge of the traditional use of plants. Informants aged < 20 and 21–40 years-old had little knowledge about the traditional uses of plants. Thirty-three plants were cited by men (34%), 22 plants by women (22.7%), and 42 species (43.3%) by both sexes.

The traditional medicinal practices of the mentioned plants are still in use in all studied districts. The majority of elderly respondents had extensive traditional knowledge of the use of plants in folk medicine. It was found that non-governmental people, such as farmers, traders, shepherds, and housewives, had more information on the ethnomedicinal uses of plants than governmental people. This is because the majority of unschooled people, particularly in rural communities, believe that traditional healing is safer than modern medicine. Moreover, it was noticed that the following wild plants: *Visnaga daucoides*, *Commiphora myrrha*, *Mentha longifolia*, *Myrtus communis*, *Ruta chalepensis*, and *Ziziphus spina-christi* were commonly sold in the local markets as medicinal herbs. In this study, more than two-thirds (67%) of the questioned inhabitants in the Al Aqiq district depend on traditional medicine alone or in combination with modern medicine. For the treatment of any health problems, most of the younger informants preferred to visit clinics and hospitals rather than use traditional healing methods. According to a previous study in Saudi Arabia [59], traditional medicine is generally practiced by middle-aged to elderly women and men who exhibited a predisposed for employing medicinal plants. In the Al Aqiq and Al Baha districts, the ethnobotanical knowledge, as well as preparation and administration methods for serious diseases, still reside with elderly people and expert herbalists. Herb preparation and administration methods, dosage forms, and dosing regimens varied by district.

A total of 97 plant species belonging to 91 genera and 44 families with their traditional uses were reported by the interviewed informants (Table 2). The recorded data revealed that the area was rich in phytodiversity and the reported plants included one gymnosperm and 96 angiosperms, and the latter included 84 dicotyledonous species and 12 monocotyledonous species. Thirty-two of the reported plants were big trees, fourteen were bushy shrubs, and fifty-three were herbs. Such species diversity in the study area could possibly be attributed to the unique climate and topographical features of the Al Baha province [25]. Among the reported plants, herbs were the most harvested growth form (64%) used by the inhabitants in Al Baha province, followed by trees (28%) and shrubs (8%). This finding agreed with the results reported by Ullah et al. [60], who revealed the dominance of herb plants in treating diseases in Saudi Arabia. In the Aljumum region, west Saudi Arabia, Qari et al. [61] found that shrubs (78%), herbs (14%), and trees (8%) predominated.

**Table 2.** Ethnobotanical data of the cited plants by the local inhabitants of Al Baha province.

| Family<br>Species<br>Local Name | Parts Used | Preparation Mode | Ethnobotanical Uses | UV |
|---|---|---|---|---|
| **Amaranthaceae** | | | | |
| *Beta vulgaris* **L.**<br>**Bangar** | Roots | Direct | Hematological disorders, CVS disorders | 0.025 |
| **Amaryllidaceae** | | | | |
| *Allium cepa* **L.**<br>**Bassal** | Bulbs | Direct | GIT disorders, respiratory disorders, endocrine disorders, immunological disorders, skin disorders, ENT disorders, ophthalmic disorders, insecticidal | 0.062 |
| * *Allium sativum* **L.**<br>**Thoum** | Bulbs | Direct | CVS disorders, hematological disorders, skin disorders | 0.037 |

**Table 2.** *Cont.*

| Family<br>Species<br>Local Name | Parts Used | Preparation Mode | Ethnobotanical Uses | UV |
|---|---|---|---|---|
| **Anacardiaceae** | | | | |
| * *Pistacia lentiscus* **L.**<br>**Mistika** | Resins | Maceration,<br>decoction | GIT disorders, respiratory disorders, ENT disorders, ophthalmic disorders, insecticidal, wound healing | 0.037 |
| **Apiaceae** | | | | |
| *Ammi majus* **L.**<br>**Khela shytania** | Fruits | Decoction | Renal disorders | 0.012 |
| *Anethum graveolens* **L.**<br>**Shabbat/Sanout** | Leaves,<br>aerial parts | Infusion,<br>direct, maceration | GIT disorders, endocrine disorders | 0.037 |
| *Coriandrum sativum* **L.**<br>**Kasbra** | Fruits, leaves | Maceration | Mental and behavioral disorders, oncological disorders, immunological disorders, dental applications | 0.012 |
| * *Carum carvi* **L.**<br>**Karawia** | Fruits | Maceration, direct | GIT disorders, endocrine disorders | 0.037 |
| * *Cuminum cyminum* **L.**<br>**Shamar** | Fruits | Decoction | GIT disorders, endocrine disorders | 0.037 |
| * *Ferula assa-foetida* **L.**<br>**Heltait** | Gum | Maceration | GIT disorders | 0.012 |
| * *Foeniculum vulgare* **Mill.**<br>**Kamoon** | Fruits | Maceration | GIT disorders, endocrine disorders, respiratory disorders, weight reduction | 0.049 |
| *Petroselinum crispum*<br>**(Mill.) Fuss**<br>**Bagdonis** | Leaves | Maceration | Renal disorders, anti-inflammatory, antibacterial | 0.025 |
| * *Pimpinella anisum* **L.**<br>**Yanson** | Fruits | Decoction,<br>maceration | GIT disorders, endocrine disorders, toxicological effects | 0.049 |
| *Visnaga daucoides* **Gaertn.**<br>**Khela bladi** | Fruits | Infusion | Renal disorders | 0.025 |
| **Apocynaceae** | | | | |
| *Adenium obesum* **Roem.**<br>**and Schult.**<br>**Adana** | Latex | Paste | Skin disorders, toxicological effects | 0.012 |
| *Calotropis procera* **(Aiton)**<br>**Dryand.**<br>**Ushar** | Latex,<br>leaves | Poultice, infusion | GIT disorders, endocrine disorders, antiparasitic, toxicological effects | 0.025 |
| *Desmidorchis*<br>*retrospiciens* **Ehrenb.**<br>**Galthy** | Stem | Direct, ash | Wound healing, toxicological effects | 0.012 |
| *Periploca aphylla* **Decne.**<br>**Suwas** | Aerial parts | Direct, poultice | GIT disorders, orthopedic disorders, dental applications | 0.025 |
| **Arecaceae** | | | | |
| *Phoenix dactylifera* **L.**<br>**Tamor** | Fruits | Direct | GIT disorders, hematological disorders, orthopedic disorders, nutrition/supplement | 0.025 |

**Table 2.** *Cont.*

| Family<br>Species<br>Local Name | Parts Used | Preparation Mode | Ethnobotanical Uses | UV |
|---|---|---|---|---|
| **Asteraceae** | | | | |
| *Artemisia abyssinica*<br>**Sch.Bip. ex A.Rich.**<br>**Birk** | Leaves | Maceration, paste | Endocrine disorders, skin disorders | 0.025 |
| *Artemisia judaica* **L.**<br>**Sheih-boethran** | Aerial parts | Infusion,<br>poultice, oil | Endocrine disorders, renal disorders<br>skin disorders, toxicological effects | 0.012 |
| *Galinsoga parviflora* **Cav.** | Leaves | Poultice | Respiratory disorders, skin disorders,<br>wound healing | 0.012 |
| * *Matricaria*<br>*chamomilla* **L.**<br>**Babong** | Flowers | Decoction,<br>maceration | Orthopedic disorders, GIT disorders,<br>endocrine disorders, respiratory disorders,<br>reproductive disorders | 0.049 |
| *Psiadia punctulata* **Vatke**<br>**Tubbag** | Leaves | Poultice | Skin disorders, wound healing | 0.012 |
| *Pulicaria undulata* **(L.)**<br>**C.A.Mey.**<br>**Arfaj, Gathgath** | Aerial parts,<br>flowers | Powder, oil | Insecticidal/repellent | 0.037 |
| * *Saussurea costus* **Falc.**<br>**Gasst** | Roots | Decoction | GIT disorders | 0.025 |
| * *Seriphidium herba-alba*<br>**(Asso) Y.R.Ling**<br>**Sheih** | Aerial parts | Maceration | GIT disorders, oncological disorders,<br>immunological disorders, nutrition,<br>insecticidal | 0.025 |
| **Brassicaceae** | | | | |
| * *Coincya tournefortii*<br>**(Gouan) Alcaraz, T.E.Díaz,**<br>**Rivas Mart.and**<br>**Sánchez-Gómez**<br>**Khardl** | Leaves, fruits | Decoction, direct,<br>oil | GIT disorders, renal disorders, skin<br>disorders | 0.037 |
| *Lepidium sativum* **L.**<br>**Rashad** | Seeds | Direct | GIT disorders, hematological disorders,<br>orthopedic disorders | 0.037 |
| *Nasturtium officinale* **R.Br.**<br>**Girgir Almaa** | Aerial parts | Maceration | Antiviral, antiparasitic | 0.012 |
| **Burseraceae** | | | | |
| *Boswellia serrata* **Roxb.**<br>**Kandr-luban** | Resins | Decoction | Respiratory disorders, orthopedic<br>disorders, smoke session | 0.037 |
| *Commiphora gileadensis*<br>**(L.) C.Chr.**<br>**Busham** | Stem,<br>twigs | Decoction, direct | Respiratory disorders, skin disorders,<br>orthopedic disorders, dental applications | 0.037 |
| *Commiphora myrrha* **Engl.**<br>**MurrHegaji** | Resins | Maceration,<br>powder, paste | GIT disorders, respiratory disorders,<br>hematological disorders, oncological<br>disorders, orthopedic disorders,<br>immunological disorders,<br>anti-inflammatory, ENT disorders,<br>ophthalmic disorders, antifungal, dental<br>applications, wound healing | 0.074 |
| **Cactaceae** | | | | |
| *Opuntia ficus*-indica **(L.)**<br>**Mill.**<br>**Barshwomi** | Fruits | Poultice, direct | GIT disorders, skin disorders | 0.025 |

**Table 2.** *Cont.*

| Family<br>Species<br>Local Name | Parts Used | Preparation Mode | Ethnobotanical Uses | UV |
|---|---|---|---|---|
| **Cannabaceae** | | | | |
| *Trema orientalis* **(L.)** **Blume** **Shubarig** | Fruits | Direct | GIT disorders, renal disorders, mental and behavioral disorders, endocrine disorders, oncological disorders, immunological disorders, nutrition/supplement | 0.012 |
| **Celastraceae** | | | | |
| *Gymnosporia heterophylla* **Loes.** **Athrara** | Twigs | Decoction | Renal disorders | 0.012 |
| **Cucurbitaceae** | | | | |
| *Citrullus colocynthis* **(L.)** **Schrad.** **Hanzal** | Fruits | Maceration | GIT disorders, respiratory disorders, orthopedic disorders | 0.025 |
| *Cucurbita maxima* **Duchesne** **Garaa** | Fruits | Direct | GIT disorders, oncological disorders, immunological disorders | 0.012 |
| *Lagenaria siceraria* **(Molina) Standl.** **Garaamurr** | Fruits | Direct | Endocrine disorders | 0.012 |
| **Cupressaceae** | | | | |
| *Juniperus procera* **Hochst.** **ex Endl.** **Arar** | Stem | Maceration, direct | GIT disorders, orthopedic disorders | 0.025 |
| **Euphorbiaceae** | | | | |
| *Ricinus communis* **L.** **Khirwee** | Leaves, fruits, seeds | Maceration, poultice, oil | GIT disorders, mental and behavioral disorders, skin disorders | 0.037 |
| *Euphorbia granulata* **Forssk.** **Umlabben** | Latex, whole plant | Maceration | GIT disorders, renal disorders, oncological disorders, immunological disorders, antibacterial, antifungal, antiparasitic | 0.012 |
| **Fabaceae** | | | | |
| *Medicago sativa* **L.** **Barsim** | Aerial parts | Decoction, direct | Endocrine disorders, hematological disorders, oncological disorders, immunological disorders, | 0.049 |
| *Senna alexandrina* **Mill.** **Sanamekki** | Leaves, fruits | Decoction | GIT disorders, | 0.037 |
| *Tamarindus indica* **L.** **Aradeeb** | Fruits | Maceration | Endocrine disorders | 0.012 |
| ***Trigonella* foenum-graecum* **L.** **Helba** | Seeds | Decoction, infusion | GIT disorders, endocrine disorders, CNS disorders, skin disorders, hematological disorders, rheumatic disorders, orthopedic disorders, antibacterial | 0.074 |
| *Vachellia nilotica* **(L.)** **P.J.H. Hurter and Mabb.** **Garz** | Fruits | Fumigation | Respiratory disorders | 0.012 |

**Table 2.** *Cont.*

| Family<br>Species<br>Local Name | Parts Used | Preparation Mode | Ethnobotanical Uses | UV |
|---|---|---|---|---|
| **Lamiaceae** | | | | |
| *Lavandula dentata* **L.**<br>**Dhrum** | Aerial parts | Poultice | GIT disorders, skin disorders, orthopedic disorders | 0.037 |
| *Mentha longifolia* **L.**<br>**Habag** | Aerial parts | Decoction, maceration | GIT disorders, CVS disorders | 0.037 |
| *Mentha spicata* **L.**<br>**Naana** | Aerial parts | Decoction, maceration | GIT disorders, respiratory disorders | 0.049 |
| *Ocimum basilicum* **L.**<br>**Rehan** | Aerial parts, leaves | Maceration, decoction, infusion | GIT disorders, respiratory disorders, endocrine disorders, CVS disorders, renal disorders | 0.062 |
| *Otostegia fruticosa* **subsp.** *schimperi* **(Benth.) Sebald**<br>**Elsharam** | Leaves | Maceration | GIT disorders, ophthalmic disorders, CNS disorders | 0.012 |
| *Plectranthus asirensis*<br>**J.R.I.Wood**<br>**Shar aseri** | Leaves | Maceration, infusion | Respiratory disorders, CVS disorders | 0.012 |
| *Plectranthus barbatus*<br>**Andrews**<br>**Shaar, Regma** | Leaves | Maceration | Respiratory disorders, ENT disorders | 0.012 |
| *Rosmarinus officinalis* **L.**<br>**Ekleel algabal** | Leaves | Infusion | GIT disorders, CNS disorders, CVS disorders, renal disorders, anti-inflammatory, supplement | 0.037 |
| *Salvia officinalis* **L.**<br>**Mermiah** | Aerial parts | Maceration | GIT disorders, mental and behavioral disorders, oncological disorders, immunological disorders, dental applications | 0.062 |
| ***\* Thymbra capitata* (L.)**<br>**Cav.**<br>**Zattar** | Aerial parts | Maceration | Respiratory disorders, GIT disorders oncological disorders, immunological disorders | 0.037 |
| **Lauraceae** | | | | |
| ***\* Cinnamomum verum***<br>**J.Presl**<br>**Girfa** | Bark | Decoction | GIT disorders, CNS disorders, endocrine disorders, hematological disorders, skin disorders, orthopedic disorders | 0.049 |
| **Lythraceae** | | | | |
| *Lawsonia inermis* **L.**<br>**Henna** | Leaves | Poultice | Mental and behavioral disorders, skin disorders | 0.049 |
| *Punica granatum* **L.**<br>**Roman** | Fruits (peels) | Direct, powder, decoction | GIT disorders, CVS disorders, hematological disorders, rheumatic disorders, oncological disorders, immunological disorders, nutrition, insecticidal, weight reduction | 0.049 |
| **Malvaceae** | | | | |
| *Malva parviflora* **L.**<br>**Khubaiza** | Leaves | Maceration | Reproductive disorders | 0.012 |
| **Meliaceae** | | | | |
| *Azadirachta indica* **A.Juss.**<br>**Neem** | Fruits, leaves | Poultice, oil | Endocrine disorders, skin disorders, toxicological effects | 0.037 |

**Table 2.** *Cont.*

| Family<br>Species<br>Local Name | Parts Used | Preparation Mode | Ethnobotanical Uses | UV |
|---|---|---|---|---|
| **Moraceae** | | | | |
| *Ficus palmata* **Forssk.**<br>**Hamadh** | Latex, fruits, leaves | Paste, direct | Endocrine disorders, skin disorders, CVS disorders, hematological disorders, immunological disorders, insecticidal, weight reduction, wound healing | 0.062 |
| **Moringaceae** | | | | |
| *Moringa oleifera* **Lam.**<br>**Moringa** | Leaves | Maceration, direct | Renal disorders, endocrine disorders, hematological disorders, dental applications, insecticidal, weight reduction, nutrition | 0.025 |
| **Myrtaceae** | | | | |
| *Eucalyptus* **sp.**<br>**Keena** | Leaves | Maceration | Respiratory disorders | 0.012 |
| *Myrtus communis* **L.**<br>**Alhdass** | Leaves | Direct, powder | Mental and behavioral disorders, wound healing | 0.012 |
| *Psidium guajava* **L.**<br>**Guava** | Fruits | Direct | GIT disorders | 0.012 |
| ***\* Syzygium aromaticum***<br>**(L.) Merr. and L.M.Perry**<br>**Gronful** | Flowers | Decoction, maceration, oil | GIT disorders, CNS disorders, respiratory disorders, orthopedic disorders, skin disorders, oncological disorders, immunological disorders, dental applications, toxicological effects, supplement | 0.062 |
| **Nitrariaceae** | | | | |
| *Peganum harmala* **L.**<br>**Harmal** | Leaves | Decoction | GIT disorders | 0.012 |
| **Oleaceae** | | | | |
| *Olea europaea* **L.**<br>**Zaytoon** | Fruits | Oil | Respiratory disorders, endocrine disorders, reproductive disorders, skin disorders, orthopedic disorders, antibacterial | 0.037 |
| *Olea europaea* **subsp.**<br>*cuspidata* **(Wall. and**<br>**G.Don) Cif.**<br>**Otom** | Aerial parts | Oil, direct | GIT disorders, endocrine disorders, oncological disorders, immunological disorders | 0.037 |
| **Pandanaceae** | | | | |
| *Pandanus*<br>*tectorius***Parkinson**<br>**Caddi** | Leaves | Maceration | Dental applications, wound healing | 0.012 |
| **Pedaliaceae** | | | | |
| ***\* Sesamum indicum* L.**<br>**Simsim** | Seeds | Oil | Respiratory disorders, hematological disorders, anti-inflammatory, antibacterial, nutrition | 0.025 |
| **Piperaceae** | | | | |
| ***\* Piper nigrum* L.**<br>**Fifil** | Fruits | Direct | Mental and behavioral disorders, GIT disorders, reproductive disorders, respiratory disorders, weight reduction | 0.025 |

**Table 2.** *Cont.*

| Family<br>Species<br>Local Name | Parts Used | Preparation Mode | Ethnobotanical Uses | UV |
|---|---|---|---|---|
| **Poaceae** | | | | |
| *Cymbopogon schoenanthus* **Spreng.**<br>**Adkhar-hmra** | Aerial parts | Maceration | GIT disorders | 0.012 |
| *Hordeum vulgare* **L.**<br>**Shaeer** | Fruits | Maceration | Renal disorders, skin disorders, dental applications, insecticidal | 0.037 |
| *Pennisetum glaucum* **R.Br.**<br>**Dukhun** | Fruits | Direct | GIT disorders, endocrine disorders, hematological disorders, orthopedic disorders, insecticidal, weight reduction | 0.025 |
| *Triticum aestivum* **L.**<br>**Gamih** | Fruits | Powder | Skin disorders | 0.012 |
| **Polygonaceae** | | | | |
| *Rumex nervosus* **Vahl**<br>**Othrub** | Leaves, Stem | Poultice, direct | Hematological disorders, skin disorders, dental applications, wound healing | 0.037 |
| **Ranunculaceae** | | | | |
| * *Nigella sativa* **L.**<br>**Alhabah al suwdaa** | Seeds | Direct | GIT disorders, CVS disorders, reproductive disorders, respiratory disorders, oncological disorders, immunological disorders | 0.037 |
| **Rhamnaceae** | | | | |
| *Ziziphus spina-christi* **(L.)**<br>**Desf.**<br>**Sedder** | Leaves, fruits | Poultice, direct, infusion | GIT disorders, hematological disorders, mental and behavioral disorders, skin disorders, antifungal | 0.062 |
| **Rosaceae** | | | | |
| *Prunus armeniaca* **L.**<br>**Mishmish** | Fruits | Direct | Respiratory disorders, skin disorders | 0.012 |
| *Rubus asirensis* **D.F.Chamb.**<br>**Akrash- Toot bari** | Fruits | Direct | Anti-inflammatory | 0.012 |
| **Rutaceae** | | | | |
| *Citrus aurantiifolia* **(Christm.) Swingle**<br>**Lemon** | Fruits | Juice | Oncological disorders, immunological disorders, antiviral | 0.012 |
| *Ruta chalepensis* **L.**<br>**Suthab** | Aerial parts | Oil | Mental and behavioral disorders, orthopedic disorders | 0.025 |
| **Salvadoracea** | | | | |
| *Salvadora persica* **L.**<br>**Arak** | Stem | Direct | Antibacterial, antifungal, dental applications | 0.025 |
| **Sapindaceae** | | | | |
| *Dodonaea viscosa* **Jacq.**<br>**Shath** | Stem, twigs | Direct | Orthopedic disorders, dental applications | 0.025 |

**Table 2.** *Cont.*

| Family<br>Species<br>Local Name | Parts Used | Preparation Mode | Ethnobotanical Uses | UV |
|---|---|---|---|---|
| **Solanaceae** | | | | |
| *Datura stramonium* **L.**<br>**Datura, Banj** | Fruits | Maceration | Mental and behavioral disorders, toxicological effects | 0.012 |
| *Solanum incanum* **L.**<br>**Hadag, aeinalbagar** | Fruits | Maceration, ash, paste | Renal disorders, anti-inflammatory, ophthalmic disorders, antibacterial, dental applications | 0.037 |
| **Tamaricaceae** | | | | |
| *Tamarix aphylla* **(L.)**<br>**H.Karst.**<br>**Athl** | Twigs | Decoction, poultice, fumigation, direct | Anti-inflammatory, dental applications, wound healing | 0.037 |
| **Urticaceae** | | | | |
| *Urtica pilulifera* **L.**<br>**Haraq** | Leaves | Poultice | Rheumatic disorders | 0.012 |
| **Vitaceae** | | | | |
| *Cissusrotundifolia* **Vahl**<br>**Ghelf** | Leaves | Direct | Orthopedic disorders, weight reduction | 0.012 |
| **Xanthorrhoeaceae** | | | | |
| *Aloe vera* **(L.) Burm.f.**<br>**Sabar** | Leaves | Poultice, paste | Skin disorders, wound healing | 0.025 |
| **Zingiberaceae** | | | | |
| ***Curcuma longa* L.**<br>**Korkom** | Rhizome | Maceration, decoction | GIT disorders, hematological disorders, oncological disorders, immunological disorders, mental and behavioral disorders, respiratory disorders, insecticidal, weight reduction | 0.049 |
| ***Elettaria cardamomum***<br>**(L.) Maton**<br>**Hail** | Fruits | Decoction, infusion | GIT disorders, CVS disorders, skin disorders, respiratory disorders, renal disorders, antibacterial, dental applications | 0.049 |
| ***Zingiber officinale***<br>**Roscoe**<br>**Zangabeel** | Rhizome | Decoction | GIT disorders, respiratory disorders, CNS disorders, hematological disorders, endocrine disorders, rheumatic disorders, orthopedic disorders, immunological disorders, antibacterial, supplement, insecticidal | 0.086 |
| **Zygophyllaceae** | | | | |
| *Tribulus terrestris* **L.**<br>**Shirshir** | Fruits | Decoction | Reproductive disorders, renal disorders, toxicological effects | 0.012 |

GIT: Gastrointestinal tract; CNS: central nervous system; CVS: cardiovascular system; ENT: ear, nose, and throat; *: Exotic plant.

The estimated family importance values (FIV) showed that Lamiaceae (0.35), Apiaceae (0.33), and Asteraceae (0.21) were considered the most important ethnomedicinal plant families in Al Baha province (Figure 2). These findings were in contrary to Aati et al. [23], who reported Asteraceae and Fabaceae as the most important ethnomedicinal families in the KSA, as well as Ali et al. [40], who reported Fabaceae and Euphorbiaceae as important ethnomedicinal families in Al Baha city and its outskirts. The study carried out by Qari et al. [61] revealed that the most-cited families in the Aljumum Region, west Saudi

Arabia were Fabaceae (32.35%), Poaceae (20.58%), Asteraceae, and Brassicaceae (17.64%). Furthermore, Asteraceae (10.48%), Lamiaceae, and Apocynaceae (7.25%) were found to be the most important ethnomedicinal families in the Jazan region, southwestern Saudi Arabia [32]. The recorded families in this study have been reported in the Kingdom's flora [24,62] and some of the species in these families have therapeutic uses [63,64].

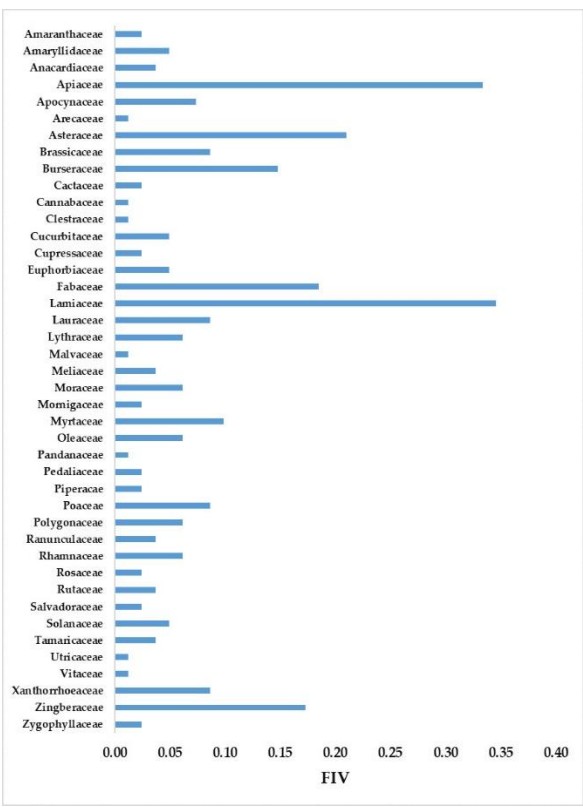

**Figure 2.** The family importance values (FIV) of the cited plants in Al Baha province.

The use value (UV) demonstrates the relative importance of the use of medicinal plants in a particular region [65,66]. As shown in Table 2, the most useful plants in Al Baha province were *Zingiber officinale* (UV = 0.086), *Commiphora myrrha,* and *Trigonella foenum-graecum* (UV = 0.074). Six of the reported plants (*Allium cepa, Ocimum basilicum, Salvia officinalis, Ficus palmate, Syzygium aromaticum,* and *Ziziphus spina-christi*) showed a moderate use value index (0.062), while the least value (0.012) was scored by thirty-four species. The use of ginger (*Zingiber officinale*) to treat a variety of disorders dates back to the spice trade and its use as a remedy for various diseases by Arab communities [67]. Tounekti et al. [32] found that *Ziziphus spina-christi, Calotropis procera,* and *Datura stramonium* had the highest range of therapeutic uses in the Jazan region. In general, spices are rich in bioactive compounds and contain distinct phytochemical ingredients, making them popular therapeutic plants [68].

The plant part used for specific treatment varies from one species to another, and from one informant to another. Fruits (30.7%) and leaves (25.4%) were the most frequently used parts of the cited plants, whereas bark, gum, and whole plant (0.9%) were the least-used parts (Figure 3). In Saudi Arabian folk medicine, the majority of herbal preparations are made from whole plants, seeds, and aerial parts of the plants [60]. These findings disagreed with previous studies in the Al Baha [40], Tabuk [36], and Aljumum [61] regions of Saudi Arabia. All these studies reported that leaves are the most-used part of the plants, which is in line with the worldwide observation [69].

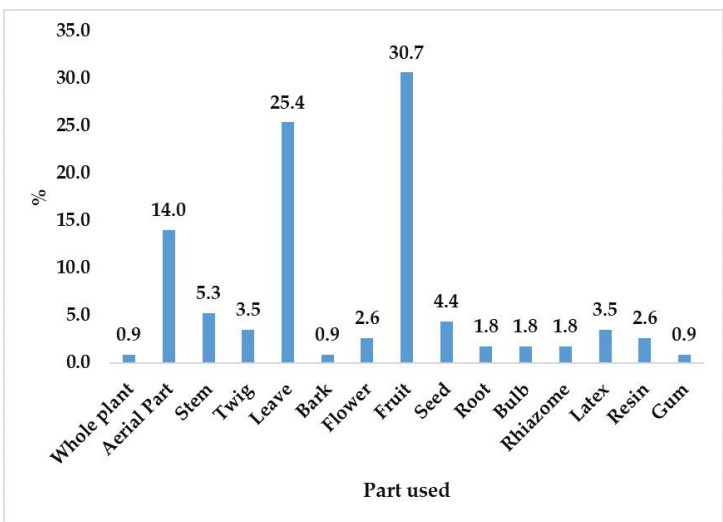

**Figure 3.** Frequency of plant parts used by inhabitants in Al Baha province.

However, using fruits and leaves in folk medicine rather than barks, roots, and whole plants is one of the most eco-friendly habits. Eleven common modes of preparation were used by the inhabitants (direct use, paste, decoction, maceration, infusion, poultice, oil, fumigation, ash, juice, and powder) to make their herbal preparations. The most preferred forms of preparation were maceration (24.7%), direct use (23.3%), and decoction (17.1%) (Figure 4). These results disagreed with the results of Ullah et al. [60], who reported that the most commonly used preparation modes in Saudi traditional medicine were decoction and infusion. Oyedeji-Amusa et al. [70], who reviewed the ethnobotany of the Meliaceae family in South Africa, revealed that decoction was the predominant mode, followed by infusion, direct use, powder, poultice, and maceration.

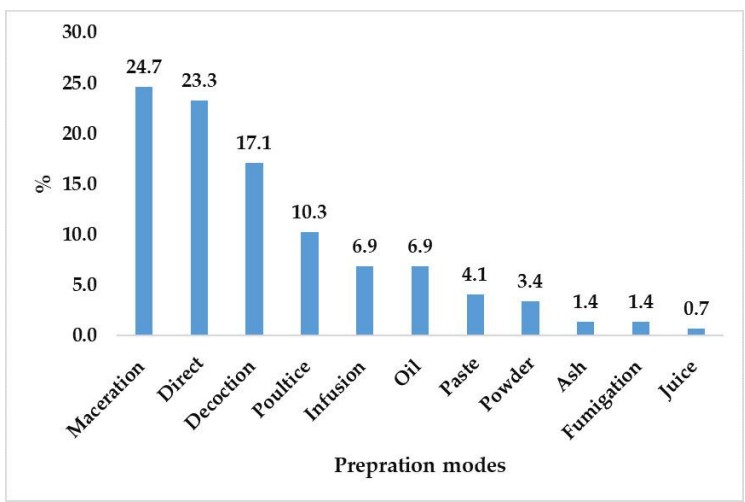

**Figure 4.** Frequency of preparation modes used by inhabitants for different ethnobotanical uses in Al Baha province.

The reported plants were either used alone or mixed with other taxa and were taken either fresh or dry. For poultice applications, the prepared herb was applied by the locals directly to the target area or covered with a clean cloth before applying. The most commonly used liquid for oral and external herbal preparations was almost always water. In some cases, for oral administration, other additives such as cold or warm milk, sour milk, butter, olive oil, and honey were mixed with the prepared herbs.

The ethnopharmacological study carried out by Ali et al. [40] on the medicinal plants of Al Baha city and its outskirts, revealed that 39 plants had medicinal benefits, and the most-used part of these plants was the leaves (49%), and paste (27.7%) was the most frequently used method for preparation.

The interviewees noted that the wild plants which were used to treat common diseases in the area were distributed in different habitats such as valleys, high mountains, and coastal lands. It was observed that the cultivated plants were obtained from nearby farmland or home gardens, and the exotic plants were purchased from the local markets or herbal shops.

Nearly half of the cited plants (49.5%) were used for the treatment of gastrointestinal tract (GIT) disorders. Percentages of the usage of plants for the other systemic disorders were 25.8, 24.7, 17.5, 17.5, 15.5, 10.3, and 6.2% for respiratory, endocrine, central nervous system/mental and behavioral, hematological, renal, cardiovascular, and reproductive disorders, respectively (Figure 5A).

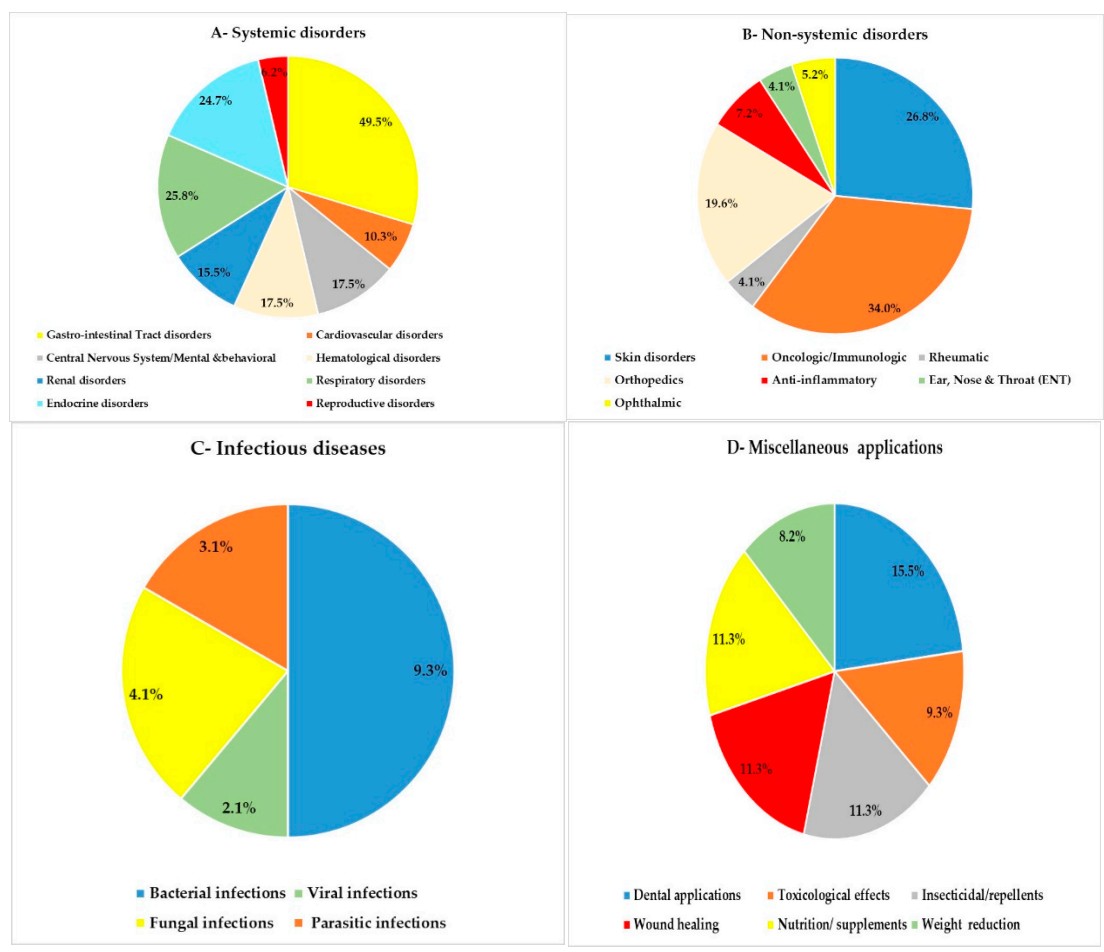

**Figure 5.** Frequency percentage of the usage of plants for various use categories in Al Baha province. (**A**) Systemic disorders; (**B**) non-systemic disorders; (**C**) infectious diseases; (**D**) miscellaneous applications.

The GIT disorders were found to be the most common diseases in the study area. In contrast, Ali et al. [40] found GIT problems as the second category, comparable to skin diseases, in Al Baha city. Alharbi [36] reported 81 plants in the Tabuk region, northwestern Saudi Arabia, that are used to treat GIT disorders. On the other hand, the ethnobotany study carried out by Hernández et al. [71] in Mexico agreed with these findings. Maceration and direct use, which were the major modes of preparation in this study, were suitable applications for the treatment of GIT disorders. However, Ribeiro et al. [72] and Jadid et al. [73] reported infusion and decoction as the most suitable herbal preparations for treating GIT diseases, respectively.

The traditional treatments of reproductive (6.2%) and cardiovascular (10.3%) disorders were found to be the least-reported systemic disorders (Figure 5A). Cardiovascular disorder is one of the most common medical problems, especially in high-altitude areas [74]. Because the Al Baha province is one of the kingdom's high regions, where heart illnesses abound, it is expected that the use of plants as a cure for heart diseases would be common. However, few (10 species) of the reported plants in the study area were utilized for cardiovascular treatment, which was unexpected in this investigation. This could be attributed to the availability and preferability of high-quality modern healthcare in Al Baha province.

For non-systemic disorders, thirty-three species (34.0%) were utilized for treating oncological/immunological disorders and twenty-six (26.8%) were used by indigenous people for skin disorders. Only four plants (4.1%) were used to treat both ENT and rheumatic disorders (Figure 5B).

Regarding the infectious category, it was observed that 9.3% of the mentioned species were used for the treatment of bacterial infections. Two plants were used for treating viral, fungal, and parasitic infections (Figure 5C).

As shown in Figure 5D, approximately 15.5% of the cited taxa were used for dental applications, followed by insecticidal/repellents, wound healing, nutrition/supplements (11.3%), and weight loss (8.2%). The use of plants for dental care, such as teeth brushing, gingivitis, mouth wash, and toothaches, was in accordance with the Islamic regulations and cultural behavior of Saudi people. This is in line with a recent trend that re-directs scientists to nature to seek a promising and interesting alternative to conventional dental therapy [75]. It was clear that significant attention was paid to the use of plants for nutritional and body weight control among the Al Baha population. The use of plants as appetite suppressants and for weight reduction was reported by Pare et al. [76].

Eleven species of the reported plants were reported as insecticidal or insect repellents. These plants could possibly be used as good sources of natural pesticides and insecticides. Souto et al. [77] recommend searching for natural secure plant-derived pesticides because synthetic pesticides and insecticides are significant global pollutants.

Several researchers have conducted ethnomedicinal studies on various communities or regions throughout the Kingdom [23,35,40,61,63]. Although many of the local people in the Al Baha province have access to high-quality modern healthcare systems, local communities still rely on plants for essential healthcare needs. They have rich ethnomedicinal knowledge about the therapeutic use of plants. However, Abulafatih [35] and Rahman et al. [63] demonstrated that Saudi people still rely heavily on folk medicine for treating various diseases. Turning back to ethnomedicine in different countries is a recent trend [78].

The agreement or disagreement level among the respondents on specific plants for the treatment of a particular use report was examined using the ICF value. Informant consensus factor (ICF) results of the reported used categories were in the range of 0.00–0.48 (Table 3). The highest ICF values in the systemic disorders category was reported for central nervous system/mental and behavioral disorders (ICF = 0.48), reproductive (ICF = 0.38), and gastrointestinal tract (ICF = 0.37) disorders. In the study carried out by Alqethami et al. [38], a high level of agreement was noticed towards the medicinal plants of Jeddah for treating respiratory and GIT disorders. This may be due to variability in informant knowledge, their areas, and the existence of the specific plant taxa in a specific area.

**Table 3.** Informant consensus factor (ICF) for various use categories.

| No. | Use Category | No. of Taxa Used (Nt) | No. of Use Reports (Nur) | ICF |
|---|---|---|---|---|
| | A- Systemic disorders | | | |
| 1 | Gastrointestinal tract disorders | 48 | 76 | 0.37 |
| 2 | Cardiovascular disorders | 10 | 14 | 0.30 |
| 3 | Central nervous system/mental and behavioral disorders | 17 | 32 | 0.48 |
| 4 | Hematological disorders | 17 | 25 | 0.33 |
| 5 | Renal disorders | 15 | 15 | 0.00 |
| 6 | Respiratory disorders | 25 | 36 | 0.31 |
| 7 | Endocrine disorders | 24 | 24 | 0.00 |
| 8 | Reproductive disorders | 6 | 9 | 0.38 |
| | B- Non-systemic disorders | | | |
| 1 | Skin disorders | 26 | 27 | 0.04 |
| 2 | Oncological/immunological disorders | 33 | 33 | 0.00 |
| 3 | Rheumatic disorders | 4 | 4 | 0.00 |
| 4 | Orthopedic disorders | 19 | 19 | 0.00 |
| 5 | Inflammatory disorders | 7 | 7 | 0.00 |
| 6 | Ear, nose, and throat disorders | 4 | 4 | 0.00 |
| 7 | Ophthalmic disorders | 5 | 6 | 0.20 |
| | C- Infectious diseases | | | |
| 1 | Bacterial infections | 9 | 24 | 0.65 |
| 2 | Viral infections | 2 | 16 | 0.93 |
| 3 | Fungal infections | 4 | 17 | 0.81 |
| 4 | Parasitic infections | 3 | 19 | 0.89 |
| | D- Miscellaneous applications | | | |
| 1 | Dental applications | 15 | 19 | 0.22 |
| 2 | Toxicological effects | 9 | 9 | 0.00 |
| 3 | Insecticidal/repellents | 11 | 11 | 0.00 |
| 4 | Wound healing | 11 | 14 | 0.23 |
| 5 | Nutrition/supplements | 11 | 14 | 0.23 |
| 6 | Weight reduction | 8 | 14 | 0.46 |

The lowest ICF values (ICF = 0.00) in the systemic disorders group category were reported for renal and endocrine disorders. This indicates a total absolute level of disagreement among the informants regarding the use of these plants for treating renal and endocrine disorders (Table 3).

For the non-systemic disorders category, the highest ICF value (ICF = 0.20) was reported for ophthalmic disorders. There was a total and absolute disagreement (ICF = 0.00) among the informants regarding the plants that were used to treat oncological/immunological, rheumatic, orthopedic, ENT, and inflammatory disorders (Table 3).

The informants showed high levels of agreement towards plants that were utilized as antiviral (ICF = 0.93), antiparasitic (ICF = 0.89), antifungal (ICF = 0.81), and antibacterial (ICF = 0.65) agents. This indicates that the local community has confidence in the plants in the study area to treat infectious diseases. There was a weak disagreement level (ICF = 0.00–0.23) among the informants regarding the plants that were used for dental applications, toxicological effects, insecticidal/repellents, wound healing, and nutri-

tion/supplements. The interviewees' level of agreement with the use of plants for weight loss was found to be modest (ICF = 0.46). As the efficacy of medicinal plants is highly correlated with ICF values, the recorded ICF values of this study could possibly be helpful for selecting plants for pharmacological research [79–81].

Table 4 shows FL, RPL, and ROP values for 42 out of 97 plants that were claimed to be used by three or more participants to treat a specific use category. The calculated FL, RPL, and ROP values for the selected plants ranged from 40–100%, 0.75–1, and 40–100, respectively. *Curcuma longa* had a very high potential (ROP = 100) for enhancing immunity. *Syzygium aromaticum* demonstrated a relatively high ability (ROP = 80) for dental applications, especially toothache. Twenty-three of the species showed moderate potential (ROP = 50) for treating the mentioned disorders. *Ocimum basilicum* and *Mentha spicata* had a relatively low potential (ROP = 40) for treating GIT disorders, especially abdominal cramps. The plants that attained low ROP values probably did so due to either modernization among the recent generations or the fact that there is a generation gap due to contemporary lifestyle changes [82]. The calculated FL values can be used for identifying preferable species by local people for curing a specific health problem [56]. The highest FL value indicates that the plant is used more frequently in the study area [83].

**Table 4.** Major uses of plants reported by three informants or more with their FL%, RPL, and ROP value.

| Species | INAUR | NUR | Primary Use | NISE | FL% | RPL | ROP |
|---|---|---|---|---|---|---|---|
| *Allium cepa* **L.** | 5 | 20 | Common cold and flu | 3 | 60.0 | 1 | 60 |
| * *Allium sativum* **L.** | 3 | 4 | Lowering cholesterol levels, skin disorders | 2 | 66.7 | 0.75 | 50 |
| *Azadirachta indica* **A.Juss.** | 3 | 11 | Skin disorders | 3 | 100.0 | 0.75 | 75 |
| *Boswellia serrata* **Roxb.** **Kandr-luban** | 3 | 5 | Cough and expectorant | 2 | 66.7 | 1 | 67 |
| * *Coincya tournefortii* **(Gouan) Alcaraz, T.E.Díaz, Rivas Mart. and Sánchez-Gómez** | 3 | 7 | Skin disorders | 2 | 66.7 | 0.75 | 50 |
| * *Carum carvi* **L.** | 3 | 5 | GIT disorders (carminative) | 2 | 66.7 | 0.75 | 50 |
| * *Cinnamomum verum* **J.Presl** | 3 | 10 | DM | 2 | 66.7 | 0.75 | 50 |
| *Commiphora gileadensis* **(L.)** **C.Chr.** | 4 | 9 | Dental applications (gingivitis, toothbrush) | 3 | 75.0 | 1 | 75 |
| *Commiphora myrrha* **Engl** | 5 | 15 | (Ophthalmic applications, wound healing) | 3 | 60.0 | 1 | 60 |
| * *Cuminum cyminum* **L.** | 3 | 5 | GIT disorders (abdominal cramps) | 3 | 100.0 | 0.75 | 75 |
| * *Curcuma longa* **L.** | 4 | 13 | Enhancing immunity | 4 | 100.0 | 1 | 100 |
| * *Elettaria cardamomum* **(L.)** **Maton** | 4 | 14 | Dental applications (toothache, mouthwash) | 3 | 75.0 | 1 | 75 |
| *Ficus palmata* **Forssk.** | 5 | 15 | Skin disorders | 3 | 60.0 | 1 | 60 |
| * *Foeniculum vulgare* **Mill.** | 4 | 10 | GIT disorders (abdominal cramps) | 2 | 50.0 | 1 | 50 |
| *Hordeum vulgare* **L.** | 3 | 10 | Renal disorders (stone) | 2 | 66.7 | 0.75 | 50 |
| *Lavandula dentata* **L.** | 3 | 8 | Infectious diseases, pain killer | 2 | 66.7 | 1 | 67 |
| *Lawsonia inermis* **L.** | 3 | 7 | Skin disorders | 3 | 100.0 | 0.75 | 75 |
| *Lepidium sativum* **L.** | 3 | 8 | Bone fracture | 2 | 66.7 | 0.75 | 50 |

**Table 4.** *Cont.*

| Species | INAUR | NUR | Primary Use | NISE | FL% | RPL | ROP |
|---|---|---|---|---|---|---|---|
| * *Matricaria chamomilla* **L.** | 4 | 14 | Carminative, sedative, arthritis | 3 | 75.0 | 1 | 75 |
| *Medicago sativa* **L.** | 4 | 6 | Hematological disorders (anaemia) | 2 | 50.0 | 1 | 50 |
| *Mentha longifolia* **L.** | 3 | 8 | GIT disorders (carminative) | 2 | 66.7 | 0.75 | 50 |
| *Mentha spicata* **L.** | 5 | 9 | GIT disorders (carminative) | 2 | 40.0 | 1 | 40 |
| * *Nigella sativa* **L.** | 3 | 7 | Enhancing immunity | 2 | 66.7 | 0.75 | 50 |
| *Ocimum basilicum* **L.** | 5 | 9 | GIT disorders (abdominal cramps) | 2 | 40.0 | 1 | 40 |
| *Olea europaea* **L.** | 3 | 11 | Arthritis | 2 | 66.7 | 0.75 | 50 |
| *Olea europaea* **subsp.** *cuspidata* **(Wall. and G.Don) Cif.** | 3 | 8 | Enhancing immunity | 2 | 66.7 | 0.75 | 50 |
| * *Pimpinella anisum* **L.** | 4 | 9 | GIT disorders (carminative) | 3 | 75.0 | 1 | 75 |
| * *Pistacia lentiscus* **L.** | 3 | 10 | Wound healing, insecticidal | 2 | 66.7 | 0.75 | 50 |
| *Pulicaria undulata* **(L.) C.A.Mey.** | 3 | 6 | Repellent | 3 | 100.0 | 0.75 | 50 |
| *Punica granatum* **L.** | 4 | 10 | Rheumatic disorders | 2 | 50.0 | 1 | 50 |
| *Ricinus communis* **L.** | 3 | 6 | GIT disorders (constipation) | 2 | 66.7 | 0.75 | 50 |
| *Rosmarinus officinalis* **L.** | 3 | 11 | GIT disorders (abdominal cramps) | 2 | 66.7 | 0.75 | 50 |
| *Rumex nervosus* **Vahl** | 3 | 8 | Dental applications (toothache, mouthwash) | 2 | 66.7 | 0.75 | 50 |
| *Salvia officinalis* **L.** | 5 | 12 | Enhancing immunity | 3 | 60.0 | 1 | 60 |
| *Senna alexandrina* **Mill** | 3 | 3 | GIT disorders (constipation) | 2 | 66.7 | 0.75 | 50 |
| *Solanum incanum* **L.** | 3 | 7 | Dental applications (decay) | 2 | 66.7 | 0.75 | 50 |
| * *Syzygium aromaticum* **(L.) Merr. and L.M.Perry** | 5 | 21 | Dental applications (toothache) | 4 | 80.0 | 1 | 80 |
| *Tamarix aphylla* **(L.) H.Karst** | 3 | 6 | Dental applications (gingivitis) | 2 | 66.7 | 0.75 | 50 |
| * *Thymbra capitata* **(L.) Cav.** | 3 | 8 | Cough and expectorant | 3 | 100.0 | 0.75 | 50 |
| * *Trigonella foenum-graecum* **L.** | 4 | 18 | DM, skin disorders | 2 | 50.0 | 1 | 50 |
| * *Zingiber officinale* **Roscoe** | 7 | 27 | GIT disorders (NV and dyspepsia), respiratory disorders (asthma, cold and cough) | 4 | 57.1 | 1 | 57 |
| *Ziziphus spina-christi* **(L.) Desf.** | 5 | 12 | Skin disorders | 3 | 60.0 | 1 | 60 |

INAUR: number of informants cited a specific plant for a specific purpose; NUR: number of use reports cited by informants for a particular plant species, usually (NUR ≥ INAD); NISE: number of informants mentioned the use of a specific plant for a particular use report. RPL: Relative popularity level; ROP: rank order priority; DM: diabetes mellitus; NV: nausea and vomiting; *: Exotic plant.

The cited species were divided into two primary categories based on RPL values, popular (RPL = 1) and unpopular (RPL = 0.75). Friedman et al. [56] state that more investigations are needed to confirm or disprove the other benefits of the unpopular group.

Plants with a 100% fidelity level, such as *Azadirachta indica* and *Lawsonia inermis*, were widely used by indigenous people to treat skin disorders, while *Cuminum cyminum* was used to treat abdominal cramps and *Curcuma longa* was used to boost immunity. These species were highly trusted by the local communities for the treatment of the mentioned disorders. Ugulu [84] stated that plants with high fidelity levels have potential economic purposes. *Lawsonia inermis*, locally known as Hena, was found to be widely used by the local population. This is most likely because Islamic medical practices have had a direct influence on the healthcare of the Saudi people (Ibn al-Qayyim, al-Tibb al-Nabawi) [85].

In the Al Baha province, *Syzygium aromaticum* (FL = 80%, RPL = 1.0), *Commiphora gileadensis*, and *Elettaria cardamomum* (FL = 75%, RPL = 1) provided the majority of the dental treatments. For teeth brushing, *Commiphora gileadensis* was widely utilized in the highlands of Al Baha province, whilst *Salvadora persica* was frequently used in the lowlands. However, Islamic teachings and prophetic medicine both encourage paying attention to hygiene and dental care. For GIT disorders, *Pimpinella anisum* was reported as a carminative, whereas *Matricaria chamomilla* was reported for carminative and sedative purposes, and to treat arthritis (FL = 75%, RBL = 1.0) (Table 4). The significance of these plants arises from their utilization to treat the main ailment problems (GIT); besides, most of them are historically categorized as spices, since Saudi Arabia was predisposed by the spice trade route [66].

In total, twenty-one exotic species were cited in this study. All of them were used to treat a wide range of ailments, and only two species (*Ferula assa-foetida* and *Saussurea costus*) had a specific therapeutic application (GIT disorders). It was observed that all the exotic plants were used to treat GIT problems except *Sesamum indicum*. Eleven of these species (52.4%) were used to treat respiratory disorders and eight (38.1%) to cure endocrine disorders. The most preferred modes of preparation for the reported exotic plants in the study area were decoction and maceration. Various parts of these plants are used, but the most frequently used part for the treatment of various health problems was the fruit. Many of these plants were frequently used in the daily food habits of the indigenous community.

A comparison of medicinal plants cited in this study with previous works in Al Baha city and its outskirts [40] and nearby areas such as Jazan [32] and Jeddah [38] revealed the documentation of fifteen species (*Rubus asirensis*, *Visnaga daucoides*, *Cymbopogon schoenanthus*, *Medicago sativa*, *Lagenaria siceraria*, *Saussurea costus*, *Euphorbia granulata*, *Cissus rotundifolia*, *Nasturtium officinale*, *Pandanus tectorius*, *Pennisetum glaucum*, *Periploca aphylla*, *Vachellia nilotica*, *Trema orientalis*, and *Ferula assa-foetida*) as medicinal plants for the first time in Al Baha province. Most of these plants were used to treat a diverse array of health problems. The present survey also demonstrated that the indigenous communities used various parts of the plants and different modes of administration for the prepared herbs. Documentation of indigenous knowledge related to these species and their therapeutic effects is useful in providing the most fundamental evidence of the therapeutic effectiveness and safety of these plants.

## 4. Conclusions

The present survey documented the traditional knowledge and the ethnobotanical uses of plants in the Al Baha province, which will be actively transferred to the younger generations. The study found 97 beneficial plants with a variety of ethnobotanical purposes. The Lamiaceae family had the greatest ethnobotanical significance, and *Zingiber officinale*, *Commiphora myrrha*, and *Trigonella foenum-graecum* were the species most utilized by the informants. The area is rich in plants that have great ethnomedicinal potential to treat various health problems. Forty-eight species can be used to treat GIT disorders and thirty-three plants have been recorded to treat oncological/immunological disorders. The obtained results could be useful for researchers to extract bioactive compounds from these plants. GIT problems were ranked as the most common ailments treated with herbal remedies in Al Baha province. Though cardiovascular system (CVS) problems are commonly reported in high-altitude areas such as Al Baha province, few plants were mentioned by the informants to treat such problems. The study revealed that maceration

and direct application were the most common suitable preparation modes, particularly for traditional treatment of the GIT problems. Informants prefer to treat abdominal cramps by using *Cuminum cyminum* (FL = 100%) and used *Pimpinella anisum* (FL = 75%) and *Mentha longifolia* (FL = 66.7%) as carminative herbs. The recorded members of the family Apiaceae were found to treat most of the GIT problems, which is in line with its ranking as the second most-important ethnobotanical family cited by the informants. Six plants had a high healing efficacy (FL = 100), twenty species were grouped as popular (RPL = 1), and twenty-two species were unpopular (RPL < 1). The study was carried out with the intention of documenting the ethnobotanical knowledge and highlighting the potential of medicinal plants in Al Baha province, KSA. The documented data demonstrated that most of the local people still depend largely on plants because of their belief that those species are more effective in the treatment of health problems. The findings of the current study may help scientists conduct additional phytochemical and pharmacological research.

## 5. Limitations of the Study

There are few limited written documents on the cultural history of the indigenous communities of Al Baha province. One potential limitation of this study is that most traditional healers are not willing to share their private information with others. Participants are sometimes willing to pass on their knowledge of medicinal plants only to their children or family members. It is difficult to collect data from the female community due to the traditional and cultural constraints of the rural communities in Al Baha province. Interviewing large numbers of informants could have resulted in more plant species being cited, as well as broader information about the use of plants in the study area.

**Author Contributions:** Conceptualization, S.A.A.-R., A.A.A. and H.A.M.; Methodology, S.A.A.-R., A.A.A., H.A.M. and A.A.A.A.; Software, S.A.Z. and A.A.E.A.; Validation, A.A.A.A. and A.A.E.A.; Formal Analysis, A.A.A. and S.A.Z.; Investigation, S.A.A.-R., H.A.M. and A.A.A.A.; Resources, S.A.A.-R. and A.A.E.A.; Data Curation, A.A.A. and S.A.Z.; Writing—Original Draft Preparation, A.A.A. and S.A.Z.; Writing—Review and Editing, A.A.A. and S.A.Z.; Visualization, S.A.A.-R., A.A.A.A., A.A.E.A. and H.A.M.; Supervision, S.A.A.-R.; Project Administration, S.A.A.-R.; Funding Acquisition, S.A.A.-R. All authors have read and agreed to the published version of the manuscript.

**Funding:** This study was funded by the Deputyship for Research & Innovation, Ministry of Education, Saudi Arabia, under grant number MOE-BU-7-2020.

**Institutional Review Board Statement:** This study complies with current international, national, and local legislation, institutional rules and ethical best practices with regards to animal experiments, clinical studies, biodiversity rights, and indigenous knowledge rights.

**Informed Consent Statement:** Verbal informed consent was obtained from each informant before voluntary enrollment in the survey.

**Data Availability Statement:** All data presented in this study are available in article.

**Acknowledgments:** The authors extend their appreciation to the Deputyship for Research & Innovation, Ministry of Education in Saudi Arabia, for funding this research work through the project number: MOE-BU-7-2020. Moreover, the authors gratefully acknowledge all informants in Al Baha province for their collaboration and the help that they gave us throughout our survey interviews.

**Conflicts of Interest:** The authors declare no conflict of interest.

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
