# Peer review of "Qualitative and Quantitative Ethnobotanical Survey in Al Baha Province, Southwestern Saudi Arabia"

_diversity, doi:10.3390/d14100867_

Round 1

Reviewer 1 Report

This is a very interesting paper and a valuable contribution to the field!

Some improvements are needed.

Line 53: define “conventional medicine”

Lines 53-56: “ 80% of the world’s population relies on traditional medical practitioners for their healthcare needs, which reflects the importance of traditional medicine, particularly in developing countries” needs a citation.

line 61: This sentence begins with “Recently,” but supporting references are very old, most more than 20 years. More recent references should be sought. 

Line 66: There are many natural products that are much more harmful (and even lethal), and many synthetic drugs that are reliable, safe, and affordable. This sentence should say “Natural products may be more….”

Line 70: “probably” should be deleted or changed to “frequently” and supported with references.

Line 73: “was achieved” should be changed to “may be achieved.”

Line 120: Should this be “group conversation”?

Line 127: “aliens” might be changed to “foreigners” or “outsiders”

Line 135” “conducted” might be changed to “consulted”

Section 2.3: To increase credibility, herbarium specimens should be sent to a recognized herbarium. See the NYBG Index Herbariorum for a list.

Line 143: “mentioned.” might be changed to “noted.”

Section 2.4 The groupings of uses is mostly good, but it is confusing that “skin disorders” would be grouped with “(G) Mental and behavioural disorders.” Consider re-defining this group, especially to match Table 3.

Lines 211-227: Clarify that these statements are findings of this current research project.

Line 232: “uneducated” should be changed to a less discriminatory term.

Line 242: “prediction” could be changed to “predisposed.”

Table 2: What is “Direct Preparation”?

Line 294: Should this be highlighted?

Line 479: Why was a written informed consent not used?

Line 504: incorrect names: should be R. Cooper and G. Nicola.

In general, citations should be from the last 5 years, especially those involving current use of plants in pharmaceutical medicines.

Information about the cultural identity and history of the people of Al Baha province is lacking. 

There should be a “Study Limitations” section.

It should be noted how study subjects were compensated for their knowledge and/or time.

If this study leads to any financial discoveries in the future, will the original holders of the Intellectual Property be compensated? 

It should be acknowledged somewhere that the people of Al Baha province hold valuable knowledge, and that sharing this knowledge with the researchers in this study was generous.

Author Response

Dear Editor and reviewer;

We would like to express our appreciation to the editor and reviewer for the thoughtful comments and helpful suggestions. We have addressed all issues indicated in the review report, and believed that the revised version can meet the journal publication requirements. Our detailed, point-by-point responses to the reviewers’ comments are given below, whereas the corresponding revisions are marked in the revised manuscript.

Reviewer 2 Report

The article entitled: "Qualitative and Quantitative Ethnobotanical Survey in Al Baha Province, Southwestern Saudi Arabia" by Sami Asir Al-Rrobai et Al. Makes a useful contribution to the knowledge of the uses of numerous species of ethnobotanical interest. The paper is well organized and is useful for the international scientific community to be able to deepen and / or confirm, with the most modern biological and chemical research investigation techniques, the various activities encountered.

Author Response

Dear Editor and reviewer;

We would like to express our appreciation to the editor and reviewer for the helpful suggestions. We believe that the revised version can meet the journal publication requirements.

Reviewer 3 Report

The manuscript is very interesting because medicinal plants are considered as a potential source of new therapeutic agents. Furthermore, this paper is well organized and clear. However, before to be accepted for publication in Diversity (MDPI), the authors should consider a couple of minor revision points:

Young informants have little or no knowledge of the use of medicinal plants in traditional medicine. Therefore, informants under the age of 40 should not be considered in these studies (subchapter 2.2, Table 1). 

Abbreviations and acronyms should be defined for the first time when they are used, throughout the abstract and again the text of manuscript. Thus, abbreviation KSA should be explained both in the Abstract and in the manuscript (line 90). Additionally, abbreviation ENT in the abstract should also be explained.

line 80 – two times “resulted in”, one should be removed.

lines 216-217 – “the inhabitants of Al Baha province have own indigenous ethnobotanical knowledge” instead of “the inhabitants of Al Baha province own indigenous ethnobotanical knowledge”

line 357 – What does it mean – “viral” ?

Reference 10 is incorrectly quoted. Please, change as follows: Cooper, R.; Nicola, G.; Natural Products Chemistry: Sources, Separations and Structures (1st ed.). CRC Press: Boca Raton, USA, 2015, 24-29. https://doi.org/10.1201/b17244.

Author Response

Dear Editor and reviewer;

We would like to express our appreciation to the editor and reviewers for the thoughtful comments and helpful suggestions. We have addressed all issues indicated in the review report, and believed that the revised version can meet the journal publication requirements. Our detailed, point-by-point responses to the reviewers’ comments are given below, whereas the corresponding revisions are marked in the revised manuscript.
